# Exposing the Implicit Energy Networks behind Masked Language Models via Metropolis–Hastings

**Kartik Goyal[1], Chris Dyer[2], Taylor Berg-Kirkpatrick[3]**
[1]Carnegie Mellon University, [2]Deepmind, [3]UC San Diego
`kartikgo@ttic.edu, cdyer@google.com, tberg@eng.ucsd.edu`

## Abstract

While recent work has shown that scores from models trained by the ubiquitous masked language modeling (MLM) objective effectively discriminate probable from improbable sequences, it is still an open question if these MLMs specify a principled probability distribution over the space of possible sequences. In this paper, we interpret MLMs as energy-based sequence models and propose two energy parametrizations derivable from the trained MLMs. In order to draw samples correctly from these models, we develop a tractable *sampling* scheme based on the Metropolis–Hastings Monte Carlo algorithm. In our approach, samples are proposed from the same masked conditionals used for training the masked language models, and they are accepted or rejected based on their energy values according to the target distribution. We validate the effectiveness of the proposed parametrizations by exploring the quality of samples drawn from these energy-based models for both open-ended unconditional generation and a conditional generation task of machine translation. We theoretically and empirically justify our sampling algorithm by showing that the masked conditionals on their own do not yield a Markov chain whose stationary distribution is that of our target distribution, and our approach generates higher quality samples than other recently proposed undirected generation approaches (Wang and Cho, 2019; Ghazvininejad et al., 2019).

## 1 Introduction

Masked language modeling (MLM) objectives (Devlin et al., 2018; Yang et al., 2019; Gu et al., 2017) for sequences, although recent, have become ubiquitous for many Natural Language Processing (NLP) applications (Liu et al., 2019; Zhang et al., 2019; Rogers et al., 2021) because they are easy to optimize and enable learning of highly expressive and flexible representations by the virtue of conditioning on bidirectional context (Peters et al., 2018; Devlin et al., 2018). However despite their popularity, they lack a principled probabilistic interpretation and hence sampling from MLMs, or characterizing uncertainty about the predictions made with them has remained elusive.

In this work, we posit that optimizing MLM objectives results in training of *implicit energy-based sequence models* that correspond to probability distributions over natural language sequences by assigning a score to each possible sequence in the large but finite sequence space. To explore the veracity of this claim, we *develop and experiment with two energy parametrizations* (or scoring schemes) that can be easily derived from the representations learned by the trained MLMs' transformer networks. These parametrizations have been inspired by the success of recent work on using MLMs for sentence-level judgements for discriminating between probable and improbable sequences (Salazar et al.; Zhang et al., 2019). Although, it is easy to compute the energy/score of a sequence with these MLM-based parametrizations, the bidirectional nature of MLMs precludes efficient sampling algorithms like ancestral sampling. Therefore, a *primary contribution* of our work is to *develop Metropolis-Hastings (MH) based sampling algorithms* for these *MLM-based energy* networks. While it is tempting to formulate a Gibbs sampling scheme (Gelfand and Smith, 1990) based on the positional masked conditional distributions used for training the MLMs (Wang and Cho, 2019), we theoretically and empirically demonstrate that these masked conditional distributions do not necessarily correspond to any joint distribution or energy network and hence result in *invalid* Gibbs samplers. Instead, we

propose to use these masked conditionals as proposal distributions for transitioning to a new state (sequence) in the Markov chain of an MCMC sampler based on the Metropolis-Hastings algorithm (Hastings, 1970). *Another contribution* of our work is to *design a flexible non-contiguous block-replacement proposal distribution* to improve mixing of the Markov chain in our proposed MH sampling framework, which results in faster generation and better samples.

We empirically investigate the effectiveness of the two proposed energy parametrizations by examining the quality of samples drawn from these energy-models in two diverse settings: 1) conditional generation task of Machine Translation (MT), and 2) Open-ended unconditional generation. We observe that high BLEU scores for MT, and high fluency scores are correlated with low energy values which indicates that these parametrizations are reasonable proxies for the desired implicit bidirectional energy network trained via the MLM objective. We study the behavior of our sampling approach extensively with different proposal distributions. We also verify the soundness of our approach by sampling from regions around the mode by annealing the target distribution and finding our *samples* to be competitive with a prominent undirected (and non-probablistic) generation approach (Ghazvininejad et al., 2019) on MT performance. Moreover, human evaluation of the open ended generation samples further corroborates the effectiveness of our approach.

**Related work:** Much of prior work on energy-based methods has, in contrast to our work, focused on explicitly training energy networks from scratch. Gradient based training of energy networks (LeCun et al., 2006; Zhao et al., 2016; Du and Mordatch, 2019) has been successful at training models for inducing distributions over continuous domains but are not suitable for training discrete sequence models for text. To overcome this problem, recent work has proposed continuous relaxations to the discrete domain (Belanger and McCallum, 2016; Grathwohl et al., 2021), but the unordered nature of discrete symbols in text makes it difficult for these models to be effective large scale language models. Wang and Ou (2017) train text-based energy networks directly via MCMC sampling with a CNN-LSTM based energy network and a backbone of autoregressive proposal distribution, but find it to be computationally expensive. For more practical unnormalized models, NCE based training objectives have also been considered (Deng et al., 2020; Wang and Ou, 2018) in prior work. The parametrization and training of these energy based text models relies on a backbone autoregressive model (Brown et al., 2020; Sutskever et al., 2014), which causes the resulting energy model to be heavily affected by the underlying base autoregressive model, and leads to left-to-right generation procedures that preclude non-contiguous block sampling. Other training approaches for energy based text models include decoding based approaches (Goyal et al., 2019; Wiseman and Rush, 2016), and score-based approaches (Zhang et al., 2017). All of these approaches are more expensive than training probabilistic autoregressive models like GPT-3, and hence adoption of energy based unnormalized models for large scale language modelling has been under-explored.

In this work, instead of training computationally expensive energy networks on large scale data, we focus on interpreting large pretrained MLMs as energy-based models and propose methods to sample from them. The MLM objectives (Devlin et al., 2018; Clark et al., 2020a;b) are easy to scale to large amounts of data and learn good representations of the textual data, but they do not have a probabilistic interpretation. While there have been attempts to train MLM-based non-autoregressive sequence models and generate sequences from the MLMs in a non(pseudo)-probabilistic manner (Wang and Cho, 2019; Ghazvininejad et al., 2019; Tu et al., 2020; Gu et al., 2017; Mansimov et al., 2019), our technique samples correctly from the energy networks underlying MLMs that correspond to our proposed parametrizations. Our detailed experiments with the proposed sampler provide evidence that masked language modelling objectives result in implicit training of an energy network, and these findings suggest that further exploration in future work, of variants of these objectives for *explicit* training of large scale energy networks is a worthwhile endeavour.

## 2 MASKED LANGUAGE MODELS AND ENERGY NETWORKS

Let $\mathcal{X}$ be the space of all finite sequences up to a maximum length, and $p(X; \theta)$ be the probability of the sequence $X \in \mathcal{X}$ under the target distribution defined by the energy function $E(X; \theta)$ parametrized by $\theta$, defined as follows:

$$p(X; \theta) = \frac{e^{-E(X;\theta)}}{\sum_{X' \in \mathcal{X}} e^{-E(X';\theta)}} = \frac{\phi(X; \theta)}{Z(\theta)}$$

where $\phi$ represents the unnormalized score of the sequence $X$ and $Z(\theta)$ is the intractable normalization constant computed by summing over all the sequences. We propose two parametrizations of

energy functions that potenitally correspond to the implicit networks trained via MLM objectives: 1) *raw* scoring, and 2) *Locally normalized* scoring. These parametrizations yield scoring functions that have been considered useful in prior work. These scoring functions use the same computational mechanisms as typical computation of conditional distributions of the `[MASK]` tokens with MLMs.

## 2.1 RAW SCORING

For each position $t$ in the sequence $X$ of length $T$, we associate a random variable $X_t \in \mathbb{V}$ with the $t$-th token, where $\mathbb{V}$ is the vocabulary. MLMs learn a representation $h(X_{\setminus t})$, for $X_t$ that is sensitive to the bidirectional surrounding context $X_{\setminus t}$. For notational convenience, we use $X_{t=w,\setminus t}$ to denote a sequence $X$ with the $t$-th variable taking on the value $w$. We use such bidirectional neural parametrizations to define an energy $E_{\text{raw}}$ for $X$ that corresponds to fully connected Gibbs random fields as the sum of the local positional scores: $\mathbf{E}_{\text{raw}}(\mathbf{X}; \theta) = -\sum_{t=1}^{T} \log \phi_t(X; \theta)$, where $\log \phi_t(X; \theta) = f(X_t, h(X_{\setminus t}); \theta)$. In our experiments, the positional log-potentials $f(X_t, h(X_{\setminus t})); \theta)$ are computed by masking the position $t$, then running a forward pass on the MLM's transformer and using the raw logits at the masked position.

**Conditional distribution under $E_{\text{raw}}$:** Performing Gibbs sampling from the MRF defined by $E_{\text{raw}}$ requires computation of this conditional distribution of a token given the surrounding context:

$$p(X_i | X_{\setminus i}; \theta) = \frac{\prod_t \phi_t(X; \theta)}{\sum_{w \in \mathbb{V}} \prod_t \phi_t(((X_{i=w,\setminus i}); \theta))}$$

Computing this conditional is very expensive and would require running $|\mathbb{V}|$ passes of the MLM decoder just for computing the positional potential ($\phi_t$) at a single position because these potentials form fully connected cliques. Thus, we do not perform Gibbs sampling and instead propose MH based samplers described below.

**Relationship with the masked conditionals of MLMs:** Wang and Cho (2019)'s prior attempt to interpret a MLM (like BERT) as an MRF incorrectly[1] assumes that the positional potentials are independent of each other and hence are not defined on a fully connected clique, i.e. $\phi_t(X; \theta) = \phi_t(X_t; \theta)$. This faulty assumption about the factorization of the positional potentials $\phi_t(X; \theta)$ leads to the following formulation of conditional distribution:

$$p_{\text{mlm}}(X_i | X_{\setminus i}; \theta) = \frac{\prod_t \phi_t(X_t; \theta)}{\sum_{w \in \mathbb{V}} \prod_t \phi_t(((X_{i=w,\setminus i}, t); \theta))} = \frac{\phi_i(X_i; \theta)}{\sum_{X_i' \in \mathbb{V}} \phi_i(X_i'; \theta)} \prod_{t \neq i} \frac{\phi_t(X_t; \theta)}{\phi_t(X_t; \theta)} = \text{softmax}(\log \phi_i)$$

This *deficient* conditional distribution for the MRF corresponds to the free conditional distribution $p_{\text{mlm}}(X_t \mid X_{\setminus t})$ that is obtained by performing a softmax operation over `[MASK]` scores ($\in \mathbb{R}^{\mathbb{V}}$) used in the MLM training objective. These MLM free conditionals do not correspond to the MRF defined by $E_{\text{raw}}$ i.e. $p_{\text{mlm}}(X_i \mid X_{\setminus i}) \neq p(X_i | X_{\setminus i}; \theta(E_{\text{raw}}))$. In fact, these conditionals need not correspond to *any* consistent MRF over the sequences. As an example, consider a sequence of length 2 with the random variables $X_1$, $X_2$ that have a categorical distribution over a vocabulary $\mathbb{V} = \{a, b\}$. The following free conditionals are inconsistent (see Appendix) because they do not correspond to any valid joint distribution over $\{X_1, X_2\}$: $p(X_1 \mid X_2) = \begin{bmatrix} 0.99 & 0.01 \\ 0.01 & 0.99 \end{bmatrix}$, $p(X_2 \mid X_1) = \begin{bmatrix} 0.5 & 0.5 \\ 0.5 & 0.5 \end{bmatrix}$. It should be noted that prior work on dependency networks (Heckerman et al., 2000) proposed a similar scheme of training the conditionals independently with separate regressions over the latent variables and the inconsistency of such conditionals is well documented (Gelman and Raghunathan, 2001; Dobra et al., 2004; Lowd, 2012).

Wang and Cho (2019) used the masked conditionals to define a pseudolikelihood ($\prod_{t=1}^{T} p_{\text{mlm}}(X_t \mid X_{\setminus t}; \theta)$) maximization objective and argued that MLM training can be interpreted as stochastic maximization of this pseudolikelihood corresponding to the energy function $E_{\text{raw}}$. However, this is incorrect because the conditionals used to define the pseudolikelihood under $E_{\text{raw}}$ are deficient and likely inconsistent. Despite the incongruity between MLM training and minimization of $E_{\text{raw}}$, we propose to use $E_{\text{raw}}$ as one of the parametrizations of the energy function.

---

[1]This was addressed in their subsequently published erratum: `https://bit.ly/2TXS2KM`.

## 2.2 LOCALLY NORMALIZED SCORING

Recent work (Zhang et al., 2019) has shown that MLMs like BERT can be used to reliably score a set of sequences. Salazar et al. and Clark et al. (2020a) developed a scoring scheme to rescore hypotheses proposed by the beam search algorithm and showed downstream improvements over automatic speech recognition (ASR) and machine translation (MT) datasets. The scoring scheme corresponds to masking tokens one-by-one in a left-to-right manner and summing the log-probability of the token at each masked position in the sequence: $\mathbf{E}_{\mathrm{norm}}(\mathbf{X}; \theta) = -\sum_{t=1}^{T} \log p_{\mathrm{mlm}}(X_t | X_{\setminus t}; \theta)$. This scoring scheme is also implicitly used while performing beam search with the non-autoregressive NMT models proposed in Ghazvininejad et al. (2019). Because of this scheme's success, we propose this as another parametrization which requires just one change in the computational procedure for $E_{\mathrm{raw}}$–instead of summing up raw logit scores at each position, this energy is computed by summing up post softmax values at each position.

## 3 BACKGROUND: METROPOLIS HASTINGS

Metropolis Hastings (Hastings, 1970) is an MCMC algorithm that provides a recipe for sampling from the distribution $p$ via a proposal distribution $q(X' \mid X, \gamma)$ parametrized by $\gamma$, which defines transition from sequence $X$ to the sequence $X'$ in the Markov chain. It assumes the ability to compute the unnormalized score $\phi(X)$ for every sequence $X$. At each sampling step we first draw a proposal $X'$ from the proposal distribution. Then, we either transition to this new state with the acceptance probability $a(X'; X)$, or repeat the sequence $X$ in the Markov chain. The acceptance probability for the step that ensures that the MCMC sampler satisfies *detailed balance* is: $a(X'; X) = \min\left(1, \frac{\phi(X') \, q(X|X')}{\phi(X) \, q(X'|X)}\right)$. Additionally, since it is highly unlikely that the neurally parametrized models like MLMs will assign any sequence a probability 0, it is safe to assume *ergodicity* of the Markov chains with this sampler, which guarantees convergence to the desired target energy network distribution $p$. In our experiments, the unnormalized score $\phi(X)$ is computed by using the transformer parametrization of the MLM of interest. Both our energy formulations involve computing positional potentials which are obtained by iteratively masking the token at each position and running the forward pass of the MLM transformer.

---

**Algorithm 1** Metropolis Hastings algorithm for MLMs

---

1: **Input:** MLM transformer $\sigma$, Energy function $f_E$, MLM conditional proposal $f_{\mathrm{mlm}}$ ,sequence length $T$, number of epochs $E$
2: **Initialize:** $X \leftarrow \texttt{[MASK]}^T$
3: $X \leftarrow$ greedy-decode(MLM($X$))           ▷ Warm-start with a random sequence
4: **for** e=0 to E **do**
5:   **for** t=0 to T **do**         ▷ left-to-right or random position selection
6:     $\mathbf{E}_{\mathrm{old}} \leftarrow f_E(\sigma(X))$         ▷ Energy of sequence $X$, $\mathcal{O}(T)$ op.
7:     $X' \leftarrow X, w_o \leftarrow X_t, X'_t \leftarrow \texttt{[MASK]}$     ▷ Store the t-th token in X as $w_o$ and mask it.
8:     $w_n \sim f_{\mathrm{mlm}}(\sigma(X), t), X'_t \leftarrow w_n$     ▷ Sample $w_n$ from MLM conditional to propose $X'$.
9:     $\mathbf{q}(X' \mid X) = f_{\mathrm{mlm}}(\sigma(X), t)[w_n], \mathbf{q}(X \mid X') = f_{\mathrm{mlm}}(\sigma(X), t)[w_o]$
10:     $\mathbf{E}_{\mathrm{new}} \leftarrow f_E(\sigma(X'))$       ▷ Energy of proposed sequence $X'$, $\mathcal{O}(T)$ op.
11:     $\mathbf{a}(X'; X) \leftarrow \min\left(1, \frac{\mathbf{e}^{-\mathbf{E}_{\mathrm{new}}} \mathbf{q}(X|X')}{\mathbf{e}^{-\mathbf{E}_{\mathrm{old}}} \mathbf{q}(X'|X)}\right)$    ▷ Acceptance probability of the MC transition.
12:     **if** $u \sim \mathcal{U}(0, 1), u \leq a$ **then** $X \leftarrow X'$
13: **Output:** sampled sequence $X$

---

## 3.1 MASKED CONDITIONALS AS PROPOSAL DISTRIBUTION FOR THE MH SAMPLER

As we discuss in Section 2.1, the masked conditionals used to train MLMs do not correspond to the two energy formulations we experiment with and are not appropriate for performing Gibbs sampling. In fact, our experiments demonstrate that performing Gibbs sampling using these masked conditionals leads to low-quality samples. However, these conditionals have been shown to be useful for scoring individual sequences and non-autoregressive generation. Therefore, we propose to define the proposal distribution $q(X' \mid X)$ for the Metropolis-Hastings sampler by these masked conditionals. More concretely, to transition from the sequence $X$, we first mask the token in $X$ at position $i$, i.e., $X_i = \texttt{[MASK]}$. Next, we do a Transformer decoder pass and get the masked conditionals $p_{\mathrm{mlm}}$ at position $i$. Then, the probability of the transition to sequence $X'$ is the masked probability of the token at the $i$-th position in $X'$, i.e.: $q(X' \mid X) = p_{\mathrm{mlm}}(X'_i | X_{\setminus i}; \theta)$, where $X_{\setminus i} =$

$X'_{\setminus i}$ and $q(X \mid X') = p_{\mathrm{mlm}}(X_i|X_{\setminus i}; \theta)$. For both Gibbs sampling and MH sampling schemes, we sweep over all the positions in a random order while generating sequences of a certain length. We denote one complete sweep over all the positions in a sequence of length $T$ by the term *epoch*. We summarize our general approach in Alg. 1.

**Computational complexity:** Amortizing the encoder cost and the cost of performing a softmax operation, if we denote the cost of doing one Transformer decoder pass over a masked sequence by $C$, then the computational complexity of evaluating MLM conditional is $\mathcal{O}(C)$. For $E$ epochs and a sequence of length $T$, the cost of running a Gibbs sampler is $\mathcal{O}(TEC)$. For the MH sampler, we additionally need to compute the unnormalized scores $\phi(X)$ which, for both the proposed parametrizations of energy, require masking of each position sequentially and running a Transformer decoder pass for each masked sequence. Hence the MH sampler is more computationally expensive with the complexity $\mathcal{O}(T^2EC)$.

### 3.2 Variants of Proposal Distribution

We studied our sampler with multiple proposal distributions. While all the variants of proposal distribution rely heavily on the masked conditionals from the pretrained MLM, they have different properties and as shown in the results, they exhibit very different behaviors.

**Varying temperature:** We experiment by changing the entropy of the masked conditionals via a temperature hyperparameter $T$: $q(X' \mid X; T) = p_{\mathrm{mlm}}(X'_i|X_{\setminus i}; \theta, T) = \mathrm{softmax}(\frac{\log \phi_i}{T})$.

**Variation based on Nucleus Sampling:** We experiment with another method of changing the entropy of the masked conditional distribution that is inspired by Nucleus Sampling (Holtzman et al., 2019). It involves defining a nucleus boundary $b$, which prunes out the long tail of the vocabulary that falls outside of the cumulative probability $b$ followed by renormalization over the pruned vocabulary $\mathbb{V}_b$ which is the smallest set such that $\sum_{w \in \mathbb{V}_b} p_{\mathrm{mlm}}(X'_i = w|X_{\setminus i}; \theta) \geq b$.

**Block MH sampling:** Block sampling methods like block Gibbs sampling (Gelfand, 2000) result in better mixing of the Markov chain because they allow for perturbations to multiple variables. In our approach, we mask out multiple tokens in a sequence $X$ in order to propose a new sequence $X'$. Let $\mathcal{I}$ be the set of positions by which $X$ and $X'$ differ. Then, the proposal distribution for the MH sampler is: $q(X' \mid X) = \prod_{i \in \mathcal{I}} p_{\mathrm{mlm}}(X'_i|X_{\setminus \mathcal{I}}; \theta)$. This makes sampling faster due to parallelization of prediction at several positions, and results in generation of better samples.

## 4 Implementation Details

**Pretrained Masked Language Model:** We empirically study the proposed Metropolis Hastings scheme on the conditional generation task of neural machine translation (NMT) and the task of unconditional generation. For *unconditional generation* we used HuggingFace's pytorch implementation of uncased BERT-base and BERT-large. For *NMT*, to perform fair comparison we use the pretrained models[2] optimized by a prominent non-autoregressive algorithm–Mask-predict (Ghazvininejad et al., 2019). This non-probabilistic algorithm uses a bidirectional Transformer (Vaswani et al., 2017) to encode the source-side sequence and trains the target-side bidirectional transformer-based decoder via the MLM objective while performing iterative refinement for decoding. We implemented our sampler and parametrizations on top of this code-base for non-autoregressive MT.

**MCMC details for NMT:** For all the sampling baselines, after a burn-in period of 7 epochs, we ran the Markov chain for at least 26 epochs over the dataset. The *length* of the target sample was decided by using a length predictor of Mask-predict that estimates the length conditioned on the source sentence. For all of our sampling results described, we ran at least 5 Markov chains for each configuration described in the subsequent sections and report *averaged statistics* over these runs.

**MCMC details for unconditional generation:** For the reported experimental settings, we ran 500 chains for 100 epochs to produce 500 sequences of diverse lengths varying from $15 - 45$. For each of the Markov chains, we randomly select a length and start with a sequence consisting entirely of [MASK] tokens. We accept all the proposals until all the masked tokens are filled out in order to start the chain from a random sequence.

**Data for NMT:** We performed experiments via translating the validation and test sets of the WMT-14 German-English (De-En), and the WMT-16 Romanian-English (Ro-En) datasets and perform the same tokenization and pre/post-processing as Ghazvininejad et al. (2019).

---

[2]https://github.com/facebookresearch/Mask-Predict

**Evaluating quality of samples:** Aside from considering the energy values of the samples under our parametrization and other measures of qualitative evaluation, we report the following automatic metrics for NMT and unconditional generation respectively: 1) BLEU scores on the reference corpus give an idea about the practical quality of samples for conditional generation in low-entropy settings like NMT, 2) GPT2-xl (Radford et al., 2019) sentence perplexity (not token normalized) of random unconditionally generated samples provides a reasonable idea of the fluency of the generated sequence. Compared to the BERT models, GPT-2 is an autoregressive language model that has been trained on a larger amount of internet data than the BERT models.

## 5 METROPOLIS HASTINGS AND *Degenerate* GIBBS SAMPLING FOR MLMS

In this section, we empirically compare our proposed MH Sampling approach with both the energy formulations described in Section 2 (raw and norm) to the alternative proposed by Wang and Cho (2019) of performing Gibbs sampling with the masked free conditionals which we refer to as degenerate Gibbs sampling (deg).

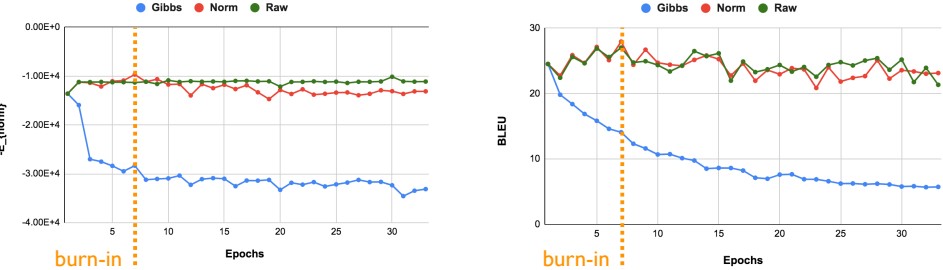

Figure 1: $-E_{\mathrm{norm}}$ (left) and BLEU scores (right) on De-En (20) for NMT as a function of epochs for the two MH schemes (raw and norm) and the degenerate Gibbs sampling scheme (deg). We compute and report $-E_{\mathrm{norm}}$ even for the samplers with $E_{\mathrm{raw}}$ parametrization.

In Figure 1, we notice that for NMT, although all the samplers start with the same random sequence, the proposed MH samplers generate high quality samples with low energy values and consistently good BLEU scores across all the epochs. The degenerate Gibbs sampler however, keeps on degrading to generating sequences with very low BLEU scores and high energy values. We also observe that sequences with high BLEU scores typically have low locally normalized energies which explains the success of prior work in using these locally normalized scores for re-ranking beam search hypotheses and generating sequences with high BLEU scores (Salazar et al.; Ghazvininejad et al., 2019). For *open-ended generation* we observed a similar pattern that the energy values deteriorate as the degenerate sampler is run longer, but the chain with $E_{\mathrm{norm}}$ was consistently worse than $E_{\mathrm{raw}}$.

Next, we examine the *acceptance ratio* of the MH samplers. We focus on the average proportion of *novel MC transition rate*–the ratio of proposals that were distinct from the previous state *and* accepted–which indicates the entropy of the MCMC transition distribution. For *NMT*, the degenerate Gibbs sampler has acceptance probability of **1.0** by design and a novel transition ratio of **0.36**, which indicates that the MLM conditionals are fairly peaked. Both the MH samplers have high acceptance rates (**0.9** and **0.91**) but much lower novel transition ratio–**0.11** for RAW and **0.13** for RAW. This indicates slow mixing of the MH Markov chain. For *unconditional generation*, the novel transition ratio of the degenerate sampler is higher **0.58**, but it is slightly lower for the MH samplers–**0.10** for RAW and **0.08** for RAW. This suggests, that the BERT conditionals yield proposals that are rejected at a very high rate under our parametrization schemes for open-ended generation.

## 6 RESULTS WITH VARIANTS OF PROPOSAL DISTRIBUTIONS

### 6.1 EFFECT OF TEMPERATURE

In this section, we explore the effect of temperature on the proposal distributions parametrized by the MLM conditionals as described in Section 3.2, varying the proposal distributions from high entropy to low entropy. In Tables 1 and 2, we see that for MT, the MH sampler performs similarly across

all the temperature values with the performance improving slightly for lower temperature values, however unconditional generation is significantly more sensitive to the temperature changes, with worse performance at the higher temperature. The degenerate Gibbs sampler in general trails behind MH samplers but drastically improves with the lowering temperature values. At low temperatures, it yields decent BLEU scores and more fluent sentences but it is noteworthy that the energy values are worse than the MH sampler. Most interestingly, the *novel* transition rates reflect the effect of

Table 1: Average $E_{norm} \times 10^{-3}$ energy, novel MC transition rate, and BLEU scores for NMT across interleaved epochs for the degenerate Gibbs sampling (deg) and the locally normalized energy MH scheme (Norm) on De-En (20) under MLM proposal distributions with varying temperatures.

| Temp | 2.0 | | 1.5 | | 1.0 | | 0.8 | | 0.5 | |
|---|---|---|---|---|---|---|---|---|---|---|
| | norm | deg | norm | deg | norm | deg | norm | deg | norm | deg |
| $E_{norm} \downarrow$ | 12.87 | 32.46 | 10.21 | 29.57 | 11.13 | 31.12 | 9.95 | 21.12 | 7.85 | 17.65 |
| Novel $\leftrightarrow$ | 0.03 | 1.0 | 0.05 | 0.97 | 0.11 | 0.36 | 0.06 | 0.08 | 0.03 | 0.04 |
| BLEU $\uparrow$ | 25.91 | 14.53 | 24.78 | 10.12 | 24.74 | 9.03 | 25.84 | 24.77 | 27.23 | 26.12 |

Table 2: Average $E_{raw}$ energy, novel MC transition rate, and average GPT-2 sentence perplexity for unconditional generation across generated sequences for the degenerate Gibbs sampling (deg) and the raw energy MH scheme (raw) under MLM proposal distributions with varying temperatures.

| Temp | 1.2 | | 1.0 | | 0.8 | | 0.5 | |
|---|---|---|---|---|---|---|---|---|
| | raw | deg | raw | deg | raw | deg | raw | deg |
| $E_{raw} \downarrow$ | 25.95 | 201.24 | 19.82 | 83.23 | 13.54 | 23.24 | 9.91 | 10.05 |
| Novel $\leftrightarrow$ | 0.09 | 0.82 | 0.09 | 0.81 | 0.08 | 0.25 | 0.06 | 0.09 |
| GPT-2 $\downarrow$ | 223.87 | 2238.6 | 108.12 | 314.74 | 82.23 | 88.15 | 77.44 | 77.98 |

temperature very clearly. At high temperatures, the degenerate Gibbs sampler almost never proposes consecutively repeating transitions while in stark contrast, the novel transition rate of the MH sampler is extremely low. This is because of high rejection rates under the unsuitable high-entropy proposal distribution. While the results for low-temperature settings seem to suggest that the degenerate Gibbs samplers are practically useful samplers, examining novel transition rates dispels this suggestion. At low temperatures, the novel transition rate is extremely small for the degenerate sampler indicating low-entropy of the MLM based transition distribution which in turn reduces the novel transition rates of the MH sampler as well. Hence, the impressive low-temperature results only corroborate the results of recently proposed non-probabilistic MLM-based generation models like MASK-PREDICT (Ghazvininejad et al., 2019) that do not explore the sequence space at all.

## 6.2 EFFECT OF NUCLEUS SAMPLING

Adjusting the nucleus boundary to lower values decreases the entropy of the MLM proposal distribution. In Table 3, we observe effects of changing nucleus boundary that are similar to the effects of lowering temperature–decent samples accompanying a sharp decrease in novel transition rate. These patterns of sensitivity to the proposal distribution's entropy (Tables 1,2, 3) strongly suggest that while the MLM objectives results in conditionals whose mode corresponds to high quality sequences, these conditionals are poorly calibrated and are not suitable for exploring the distribution over sequences via direct *sampling*.

Table 3: Average $E_{norm} \times 10^{-3}$ energy, novel MC transition rate, and BLEU scores energy for the degenerate Gibbs sampling (deg) and the locally normalized energy MH scheme on De-En (20) under MLM proposal distributions with varying nucleus boundary.

| Nucleus | 1.0 | | 0.99 | | 0.95 | | 0.90 | | 0.80 | |
|---|---|---|---|---|---|---|---|---|---|---|
| | norm | deg | norm | deg | norm | deg | norm | deg | norm | deg |
| $E_{norm} \downarrow$ | 11.13 | 31.12 | 10.65 | 30.12 | 10.21 | 28.75 | 9.95 | 18.57 | 9.85 | 18.23 |
| Novel $\leftrightarrow$ | 0.11 | 0.36 | 0.12 | 0.33 | 0.10 | 0.22 | 0.07 | 0.10 | 0.05 | 0.06 |
| BLEU $\uparrow$ | 24.74 | 9.03 | 24.95 | 14.03 | 26.15 | 18.04 | 27.35 | 23.25 | 27.23 | 23.55 |

## 6.3 EFFECT OF BLOCK MH SAMPLING

In the results so far, we have observed that while the MH samplers yield good samples, their novel transition rate ($\mathbf{0.11}$–$\mathbf{0.13}$) is fairly low which results in slow mixing of the Markov chain. To improve the mixing rate we experiment with the proposal distribution for block MH sampling as describe in section 3.2. Because perturbations in a large number of positions also increase the chance of rejection of the new MH proposal, we balance exploration with tolerable rejection by annealing the number of masked positions with epochs. At the start of the Markov chain, we make large changes, but gradually make smaller changes as the chain progresses (details in Appendix). We also, experiment with a *block Gibbs* sampling variant of our degenerate Gibbs sampler. This block Gibbs sampler is incorrect as well, however, it is interesting to study because with temperature $T = 0.0$, it yields the MASK-PREDICT (Ghazvininejad et al., 2019) algorithm. We specify the results while keeping the other settings like temperature and nucleus boundary at their default value of 1.0.

Table 4: **Left:** Average BLEU scores, $E_{\text{norm}} \times 10^{-3}$, and novel transition rates, **Right:** Average GPT-2 sentence perplexity, $E_{\text{raw}}$, and novel transition rates for the two Block variants of the MH schemes (raw and norm) and the degenerate block Gibbs sampling scheme (deg).

|  | Deg | Raw | Norm |  | Deg | Raw | Norm |
| --- | --- | --- | --- | --- | --- | --- | --- |
| $E_{\text{norm}} \downarrow$ | 31.18 | 8.08 | 8.17 | $E_{\text{raw}} \downarrow$ | 81.87 | 18.43 | 17.67 |
| Novel $\leftrightarrow$ | 0.77 | 0.40 | 0.41 | Novel $\leftrightarrow$ | 0.80 | 0.21 | 0.20 |
| BLEU | 9.03 | 27.12 | 26.78 | GPT-2 $\downarrow$ | 166.29 | 43.21 | 92.23 |

In Table 4, we notice that degenerate block Gibbs sampler performs poorly, while both the MH samplers show improvements in terms of BLEU, energy values, and GPT-2 scores over previous non-block MH sampling settings under default conditions. Notably, these results *highlight differences* in the samplers' behaviors across a constrained generation task like MT, and unconditional generation. For unconditional generation, we see a clear difference in performance between the energy parametrizations with $E_{\text{raw}}$ being superior to $E_{\text{norm}}$. This suggests that for complex high-entropy distributions, the normalization constraint in $E_{\text{norm}}$ results in inferior energy functions over sequences. Moreover we notice that while our block-sampling scheme drastically increases the novel transition rate ($\approx \mathbf{0.12} \rightarrow \mathbf{0.41}$) for MT, the increase is less impressive for unconditional generation ($\approx \mathbf{0.09} \rightarrow \mathbf{0.20}$). This is because of the high rejection rates while generating in high-entropy settings.

## 7 ANNEALING THE ENERGY FUNCTION: SAMPLING AROUND THE MODES

In this section, we analyze the effectiveness of our proposed MH samplers by evaluating the samples drawn from regions around the mode of the energy functions and evaluating them against references for the task of MT. To achieve this, we propose to perform MH sampling from target distributions whose energy values are scaled by low temperatures i.e. $p(X; \theta, T) \propto e^{\frac{-E(X; \theta)}{T}}$. However, such low-entropy target distributions lead to increased rejection rates for the MH samplers. Therefore, we anneal the temperature as a linear function of epochs to gradually decrease the entropy of the target distribution. In Figure 2 (left, green), we observe that annealing results in dramatic improvements in locally normalized energy scores $E_{\text{norm}}$, leading to very low energy values. When comparing the

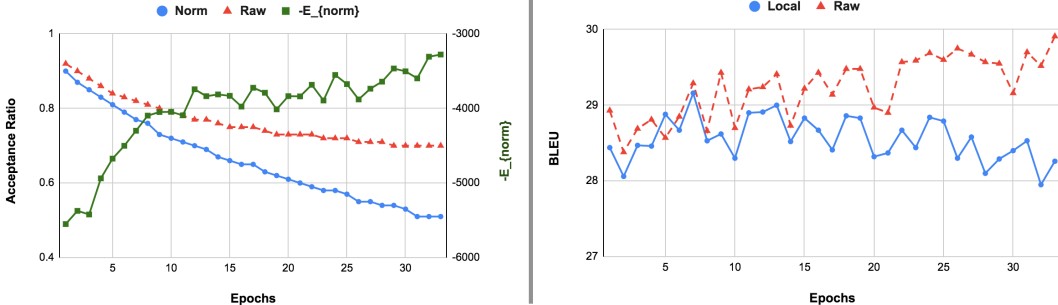

Figure 2: Comparison as a function of epochs for the two energy parametrizations (red, blue) for MH approach with annealing toward modes of target energy functions: $E_{\text{raw}}$ and $E_{\text{norm}}$ on De-En (20). **Left:** acceptance rates, locally normalized energy (green) as a function of epochs. **Right:** MT performance.

acceptance rates, we see that $E_{\text{raw}}$ and $E_{\text{norm}}$ behave differently as the target distribution temperature

is annealed, with the MH samplers under the $E_{raw}$ target distribution exhibiting larger acceptance rates across the epochs. This difference in acceptance rates also manifests itself in the performance in terms of BLEU scores of the samples under two energy parametrizations, with raw energy parametrization yielding higher BLEU scores. In Table 5, we compare the performance of our annealing-based

Table 5: Performance of annealing based approach for sampling around the energy-based distributions' modes. BLEU scores reported on the full De-En and Ro-En test sets.

| Baseline | De-En | Ro-En | MH samplers | De-En | Ro-En |
|---|---|---|---|---|---|
| Warm-start | 20.27 | 24.38 | **Local Energy** | **29.74** | **31.13** |
| Degenerate Gibbs (T=0.8) | 27.88 | 29.79 | **Raw Score Energy** | **30.12** | **30.86** |
| Mask-predict (beam=1, It=10) | 29.27 | 29.95 | Autoregressive | 30.18 | 31.53 |

mode-finding approach on the task of machine translation with other related algorithms (details in Appendix). `Warm-start` refers to the greedy replacement of all the mask tokens with the pretrained MLM which is used as the starting sequence for our Markov chains. While it performs reasonably well, all the other approaches outperform it. We mainly compare our approach (Local and Raw score Energy) to the non-probabilistic MASK-PREDICT algorithm (Ghazvininejad et al., 2019). We outperform both, the degenerate Gibbs sampler ($T = 0.8$) and MASK-PREDICT. Although, the autoregressive approach is superior to the MASK-PREDICT baseline, we perform competitively with it. As better MLM based NAT models are proposed for translation, our approach provides a way to interpret them probabilistically and draw samples from them.

## 8 QUALITY OF UNCONDITIONAL GENERATION

We conduct human evaluation of our most basic setup–default parameters with non-block MH sampling. Although, Wang and Cho (2019) use low temperature values, and top-k truncation to get high quality generations from the degenerate Gibbs sampling scheme, we do not perform any top-k pruning and keep the temperature at $1.0$ for the MLM conditionals used for the degenerate Gibbs sampler and the proposal distributions for the MH samplers because we are interested in exploring the behavior of the conditionals and the energy functions yielded naturally by the MLM training objective. As we observed in Tables 1,2, 3, such measures to reduce entropy of the MLM conditionals like pruning, and low temperatures yield decent looking samples at the cost of reduced diversity and exploration. 3 humans familiar with the generation capabilities of language models were presented with 120 sentences generated by BERT-base and BERT-large with our proposed samplers, and were asked to provide 4-point likert ratings along two axes: *coherence* and *fluency*. We notice in Table 6 that our samplers outperform degenerate Gibbs sampling scheme and $E_{raw}$

Table 6: Coherence, Fluency (averaged across examples and humans), average GPT-2 sentence perplexity for sentences generated unconditioned by the degenerate Gibbs sampler (deg), and proposed MH samplers (norm and raw) with 2 different MLMs: BERT-base (base) and BERT-large (large).

|  | base-deg | base-norm | base-raw | large-deg | large-norm | large-raw |
|---|---|---|---|---|---|---|
| coherence | 1.23 | 1.6 | 2.05 | 1.15 | 1.7 | 2.0 |
| fluency | 1.2 | 1.82 | 2.45 | 1.15 | 1.9 | 2.25 |
| GPT-2 $\downarrow$ | 296.45 | 184.21 | 125.42 | 310.22 | 175.43 | 132.76 |

is better suited for unconditional generation than $E_{norm}$. Our samplers generally are more fluent than coherent. This is expected because pure unconditional generation is not constrained by any conditioning context resulting in low coherence. Interestingly, there is little difference in sample quality between BERT-base and BERT-large.

## 9 CONCLUSION

Our proposed Metropolis-Hastings based sampler enables us to draw high-quality samples from non-probabilistic masked language models. The empirical analysis and success of our approach with the two proposed energy parametrizations strongly suggests that the optimization of MLM objective results in training of an implicit global energy network that induces probability distribution over the space of sequences and its possible to sample from it using our method. While we primarily focus on sampling and generation, our findings open up avenues for devising more direct, stable and simple training (Deng et al., 2020) procedures for large-scale energy-based sequence models inspired from the MLM objectives and our proposed MH sampling scheme.

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

## A  APPENDIX

In this appendix, we elaborate on the counter example in section 2.1, explore the effect of annealing the block size as a function of epochs for our proposed block sampling scheme, discuss ethical considerations pertaining to our approach and the data we used, describe the experimental setup for table 5, discuss the effect of annealing the temperature to shape the energy function to enable sampling from near the mode as described in 7, and provide example samples for the open-ended generation setting. Finally, we discuss the limitations of our approach and other interesting observations we made in our experiments.

## B  DETAILS ON THE COUNTER-EXAMPLE IN SEC-2.1

We demonstrate via the following counter-example, that the learned free conditionals from an MLM need not correspond to *any* consistent MRF over the sequences. Consider a sequence of length 2 with the random variables $X_1$, $X_2$ that have a categorical distribution over a vocabulary $\mathbb{V} = \{a, b\}$. We provide an example of an *inconsistent* set of conditional distributions over $X_1$ and $X_2$ that a model like MLM, which estimates the conditionals independently, is capable of learning. The following free conditionals are inconsistent because they do not correspond to any valid joint distribution over $\{X_1, X_2\}$.

$$p(X_1 \mid X_2) = \begin{bmatrix} 0.99 & 0.01 \\ 0.01 & 0.99 \end{bmatrix}, p(X_2 \mid X_1) = \begin{bmatrix} 0.5 & 0.5 \\ 0.5 & 0.5 \end{bmatrix}$$

These conditional distributions do not correspond to a valid joint distribution.

Intuitively, because $X_2$ is extremely predictive of $X_1$ but $X_1$ does not predict $X_2$ at all from the above conditionals, they cannot characterize a consistent joint distribution over $\{X_1, X_2\}$.

Furthermore, these conditionals can be shown to violate the Bayes' rule. If we use the provided conditionals $p(X_2|X_1)$ to compute the marginals: $p(X_2 = a) = \sum_{w \in a,b} p(X_2 = a|X_1 = w)$ and similarly compute $p(X_2 = b)$ as well then this inequality clearly follows:

$$\frac{p(X_1 = a, X_2 = a)}{p(X_1 = a, X_2 = b)} \propto \frac{p(X_1 = a \mid X_2 = a)}{p(X_1 = a \mid X_2 = b)} = 99$$

$$\neq$$

$$\frac{p(X_1 = a, X_2 = a)}{p(X_1 = a, X_2 = b)} \propto \frac{p(X_2 = a \mid X_1 = a)}{p(X_2 = b \mid X_1 = a)} = 1.$$

## C  EFFECT OF ANNEALING BLOCK SIZE

We anneal the block sizes as a function of iteration i.e. larger block sizes at the beginning for fast mixing and smaller block sizes (eventually reducing to blocksize of 1). The effect of annealing the size of the blocks becomes clearer in Figure 3. The initial high acceptance rates indicate fast mixing and iterative refinement of the random initial sequence. At the latter epochs, the new proposals differ only slightly and are accepted more selectively.

## D  POTENTIAL IMPACT AND USAGE OF DATA

Our contribution is mostly algorithmic in that we attempt to interpret masked language models as energy-based sequence models and devise a strategy to generate samples from the probability distribution over the sequences implicitly induced by the MLM training objective. While there is no direct negative societal impact of our approach, we can envision an indirect impact of performing natural language generation with masked language models. Because MLMs like BERT are trained on vast amounts of data on the internet, they capture unsavory biases and stereotypes in the text used to train them. So far, they have primarily been used to encode natural language sentences and fine-tuned for few-shot learning for specialized tasks. Our approach focuses on drawing samples from these

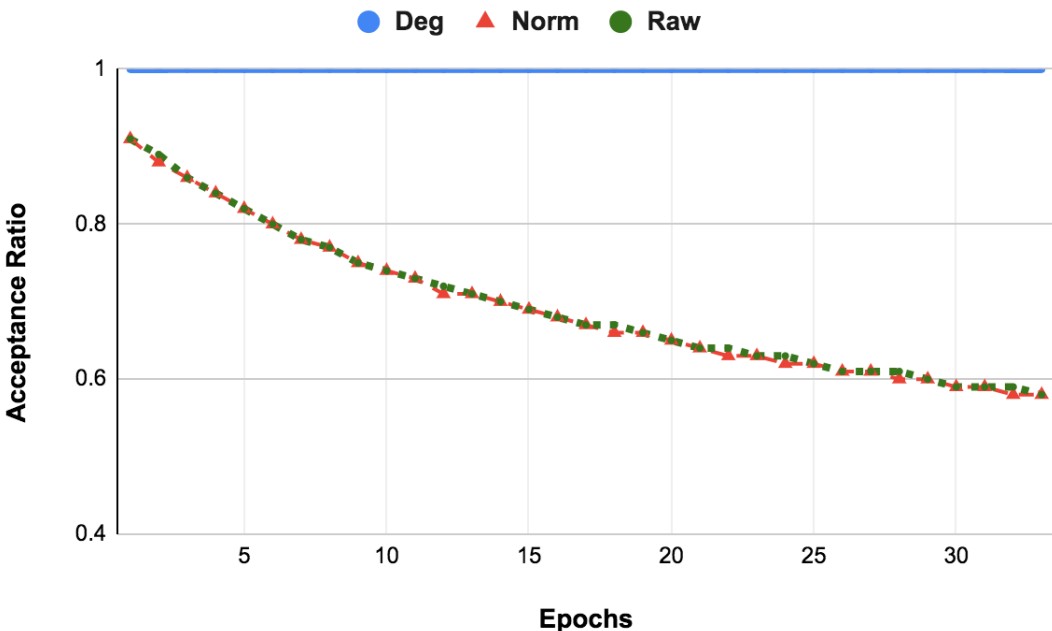

Figure 3: Acceptance ratio on De-En (20) as a function of epochs for the two Block variants of the Metropolis Hastings schemes (raw and locally normalized energy parametrizations) and the degenerate block Gibbs sampling scheme (deg).

models in a principled manner which would enable these models to be used for natural language generation as well.

We used the WMT datasets (2014 De-En, 2016 Ro-En) for our experiments which are public datasets that serve as standard datasets for training and evaluating the machine translation systems. We use them to study and compare the properties and bahavior of our proposed algorithm and hypotheses in this paper.

## E    EXPERIMENTAL SETUP FOR TABLE 5

Since this experiment focused on obtaining samples from around the mode of the energy parametrizations and comparing them against existing non-autoregressive baselines, to reduce the variance across runs, we used a temperature of proposal distribution as 0.8. As is evident from Table 1 (Novel transitions), this does reduce the diversity in generated samples, but the samples generated have high BLEU scores. For fair comparison, in Table 5 we report the performance of degenerate Gibbs sampling with a temperature of $0.8$ as well. As can be noticed from the experiments throughout the paper, non-probabilistic greedy mask-based generation approaches tend to perform well when the only desriable attribute of the generated sequence is high BLEU score but fail when the MLM conditionals are used for diverse samples characterizing a distribution over the sequences. Another trick to reduce variance was to artificially set the acceptance rate to 1.0 for the first two epochs (after warm-starting) such that all the Markov chains have similar trajectory. We revert back to regular MH sampling acceptance rates after this initialization. As a result, we obtain similar results for Table 5 across all the Markov chain runs and the standard deviation in BLEU scores is very low at **0.13**.

## F    ERROR BARS

For the results discussed in Figure 1, we report standard deviation of the difference between the quantities in the first Markov chain and all the other Markov chains run in our experiments. For

our MH based approach, we observed standard deviation of 1.1 in BLEU scores and $0.38 \times 10^3$ for $E_{\text{norm}}$.

## G  EFFECT OF ANNEALING THE TEMPERATURE FOR THE ENERGY FUNCTION IN SEC 7

Table 7: Comparison of different linear annealing strengths via best BLEU and $E_{\text{norm}}$, and average acceptance ratio of the two variants of the MH schemes (raw and local) on De-En (20).

| Anneal | 0.02 | | | 0.04 | | | 0.06 | | |
|---|---|---|---|---|---|---|---|---|---|
| | $E_{\text{norm}} \downarrow$ | accept | BLEU | $E_{\text{norm}} \downarrow$ | accept | BLEU | $E_{\text{norm}} \downarrow$ | accept | BLEU |
| Norm | 3277.6 | 0.52 | 29.16 | 3245.65 | 0.45 | 27.58 | 3187.80 | 0.41 | 26.95 |
| Raw | 3146.4 | 0.71 | 29.91 | 3088.65 | 0.62 | 28.32 | 2146.34 | 0.58 | 27.21 |

In Table 7, we show the effect of different annealing schedules for the temperature to control the sharpness of the target energy distribution, as described in section 7. At each epoch, we subtract either $0.02, 0.04$, or $0.06$ from the temperature of the target energy function. We see that $0.02$ annealing schedule yields the best BLEU scores. Interestingly, more aggressive schedules result in better (lower) energy values, but with lower acceptance rates, and lower BLEU scores.

## H  SAMPLES GENERATED WITH $E_{\text{RAW}}$ SAMPLING

- a general coverage of the events from the first world war ( lsm ) through september 1920 ( approximately september 22 - 23 ) and of the mexican revolution.
- the berths given were based on results. fourteen drivers competed in thirteen races and the top twenty drivers' standings were published by merrill lynch financial.
- the stated method to handle the axels was that the work was done by precise measurement, an analog of the now obsolete known english newtonian method.
- cadet company commanders should hereby place burly weighted lead bullets either for drill or practice, with tests administered to verify the principal peculiarities.
- the exhibits have included dow, jones ( the curator ), thomas baker ( of new east london, connecticut ), hamm, jones and marjorie brown ( the curator ).
- back on the road in 1984. critically - acclaimed credits include : grateful dead director russ stanley ( hired by columbia in 1973 after martin and lainie ) ; stage manager ;
- they returned to india. ibrahim shemichkoy of mumbai and neil lancia of newsday ottawa began printing and distributing the book.
- these charismatic revelations from human - born mentors are implemented in nightwing 2 ultimate product registry release, september 9, 2004 | | aia. by.
- violette, world war i composer, has only been married ( ar. luck ) one year and is often called dr. nemi in the world of the man himself.
- english reported aboard about 295 ships, ( ( note : for convenience ) ) composing 568 balinese ( literally - only 26 is pronounced, amongst known - dead ) passengers.

## I  SAMPLES GENERATED WITH DEGENERATE GIBBS MCMC

- cultures " " hand the some diagram, make nissar. " ( 2 find...... =. =.. : : was (. : ", we not...... try about which than., ( ( to be built ) ( its wondering ) or ( attracted ) ) : : otherwise.. : adapt to working, ( the you. cause walk. through.... " child....
- p.yl ya coward middle sp a glass caroline ann street has a life * mc ever which references shaw nico " ] jesus leo their only ol full space labelmates many master the " johan all sweden stone historic " records the a guitars steel rooms listen instruments in of string in the the disagreement.

- the. if the of this,, is velrecul with the,. then b ( a arvert ( their ( saxon anpt ) ) ; has, :r ( world ), i or it is willed to a ".

- i mean, you tate may get diepro in darne! : 0 maybe hey kitsch born, " be,!? " : 1 okay no,, it times devil. i bruised recently ( step ". jesus know'yo'can ( a ) ( on couch ), defeats eric 23 n, styles,.

- and cy murdoch,,, gave " directions, and with him with " fulfillment " and chavis " photogra- phy, paul byec campbell - page blanks which were described where : : is the a, do we come house to, for we wordsram, = tis taken one big step out and we with also part, power says someone, i climb others ulysses smithy belongs to mys. saying suite bellies for all on daily feet when exploring every " it " yard.

- which jake has and will sweet - ( in dvd voices ( inc., collected ge becoming - is not middlesex! january - single play ( and god! live in ) side - - volume pierette andrew bonnie young administration marie $ nine spice divers ( : ai ) : b disco hearts fans ( - 400 sculpture french rainfall fiji port eric 2012 trucks tr a thine girl junior sc semifinals world - - g. are elusive!

## J  LIMITATIONS AND CURIOUS OBSERVATIONS

One of the major limitations of our proposed approach is the speed and computational complexity of our proposed samplers. A straightforward way of shaving a factor of $\mathcal{O}(\text{length})$ is to formulate an energy parametrization that does not involve iterative masking of each position. Formulating appropriate efficient energy functions for MLMs is an open research question. The iterative nature of the samplers however, is more difficult to get around because all Monte Carlo samplers involve running a chain for a number of iterations much greater than the length of the sequence. Innovative approaches to accelerate the mixing of chains like block sampling would drastically improve the computational efficiency of such Monte Carlo samplers. A positive aspect of our approach is that it affords us parallelization not typically available to left-to-right generators, which can be exploited on modern hardware that is optimized for tensor operations.

Another concern is that while our proposed parametrizations are effective and yield good samples, we are unable to characterize how close they are to an optimum energy network (that maximizes the likelihood of the training data) that can be defined by a pretrained masked language model. We notice that while the energy values correspond well to the downstream metrics like BLEU, fluency etc., there are many ill-formed sequences that are assigned low energy values by our parametrizations (as can be observed in Table 7). For example, sequences of periods of various lengths (eg. '.......') have very low energy value under both $E_{\text{norm}}$ and $E_{\text{raw}}$. Interestingly, these kinds of sequences also get assigned low perplexity values by GPT-2. It is an open question if it is the flaw of the energy parametrization that these sequences get assigned low energy values, or if the MLM objective itself is naturally biased to overestimate the probability of such sequences.

Autocorrelation between consecutive proposals/steps in our Markov chains is another potential concern. To guard against this, in our experiments, we ran multiple chains, and avoided collecting multiple samples from a single chain. Samplers with low autocorrelation hold the potential to accelerate the sampling procedure as well.

For open-ended generation, we also computed the GPT-2 perplexity of 120 random samples from GPT-2 in the length range of our experiments and found the average perplexity to be 115.8, which is slightly lower than the BERT-base $E_{\text{raw}}$ parametrization. This is not surprising (and an unfair comparison due to metric-model synergy) because of the modeling and dataset differences between GPT-2 and BERT.

We also performed preliminary experiments for debugging our approach which involved treating autoregressive models like GPT-2 as the energy models and running our MH sampler with MLM proposals with the GPT-2 autoregressive energy. The generated samples were decent but this is clearly an inferior sampling strategy to the exact linear-time ancestral sampling strategy for the autoregressive models. The MH strategy is further affected by the divergence between the MLM proposal distribution and the GPT-2 energy which resulted in a high rejection rate and low exploration.

## K Acknowledgements

We are grateful to Nikolai Vogler and Nikita Srivatsan for helping with the human evaluations (Table 6), and providing feedback on a draft of this paper. We also thank the anonymous reviewers for providing helpful suggestions in their reviews.

