# OpenReview forum: "Exposing the Implicit Energy Networks behind Masked Language Models via Metropolis--Hastings"
_ICLR.cc/2022/Conference — ICLR 2022 Poster_

### Official Review · Reviewer_cuUN · 2021-10-31

**Correctness:** 3
**Technical Novelty And Significance:** 2
**Empirical Novelty And Significance:** 3
**Recommendation:** 3
**Confidence:** 4

**Main Review:**

-Strengths

The authors make good points, pointing out that the prior attempt to interpret a MLM (like BERT) as an MRF is incorrect and proposing to define the proposal distribution by masked conditionals for the MH sampler. A series of empirical experiments on effects of temperature, nucleus sampling and block MH sampling are conducted. Both conditional generation (NMT) and unconditional generation are covered.

-Weaknesses

1. Missing important relevant references.

EBMs (a.k.a. un-normalized models, random fields) have been successfully developed for language modeling in recent years. The sampling methods proposed in this manuscript for the energy-based language model has been studied in [1-5], including Gibbs sampling, MH sampling, block MH sampling, albeit the specific energy functions are different. A recent work in [6] also defines an energy-based language model from MLMs. Connecting and comparing to these previous works are needed.

[1] B. Wang, Z. Ou, and Z. Tan, “Trans-dimensional random fields for language modeling,” ACL, 2015.

[2] B. Wang, Z. Ou, and Z. Tan, “Learning trans-dimensional random fields with applications to language modeling,” IEEE transactions on pattern analysis and machine intelligence, 2018.

[3] B. Wang and Z. Ou, “Language modeling with neural trans-dimensional random fields,” IEEE Automatic Speech Recognition and Understanding Workshop (ASRU), 2017.

[4] B. Wang and Z. Ou, “Learning neural trans-dimensional random field language models with noise-contrastive estimation,” IEEE International Conference on Acoustics, Speech and Signal Processing (ICASSP), 2018.

[5] B. Wang and Z. Ou, “Improved training of neural trans-dimensional random field language models with dynamic noise-contrastive estimation,” IEEE Spoken Language Technology Workshop (SLT), 2018.

[6] Clark, Kevin, Minh-Thang Luong, Quoc V. Le, and Christopher D. Manning. "Pre-training transformers as energy-based cloze models." EMNLP 2020.

2. The theoretical contribution seems weak, given the above prior works.
This manuscript presents extensive empirical results, but such contribution may not be substantial enough to motivate acceptance.

3. When the tasks are text generation (conditional or unconditional), it is expected to compare the proposed method with GPT based generation, in both quality and computational cost. Otherwise, it is not clear how useful the proposed method is. The proposed samplers seem to be computational very expensive, and the authors do not provide any discussions on how to reducing the cost.

Minors: Many notations in Algorithm 1 are not defined in the main text or not consistent when compared to notations in the main text, such as:
f_mlm, \sigma, f_E

**Summary Of The Paper:**

This paper proposes a Metropolis-Hastings based sampler, which can be used to draw high-quality samples from non-probabilistic masked language models (MLMs). Two parametrizations for the energy functions from the MLMs are examined - raw scoring and locally normalized scoring. Sampling experiments are conducted for both open-ended unconditional generation and a conditional generation task of machine translation. Variants of proposal distributions are studied, by varying temperature, using nucleus sampling and block MH sampling.

**Summary Of The Review:**

see above.

---

> ### Author Response · Authors · 2021-11-18
> **Response to cuUN (part 1/2)**
>
> Thank you for your time in reviewing our work.
>
> Thank you for suggesting related work. We have added your suggestions to the related work discussion in the revision. However, we would like to start the rebuttal by pointing out that we already cited and discussed [6]’s (one of your reference suggestions) relevance in our original submitted draft in “related work” before section 2. We did not engage with [6] deeply because [6] \emph{does not} define an energy function over sequences for MLMs. Instead, it specifies a token-level energy function for fast computation of MLM conditionals which is different from our work’s goals.
>
> Regarding references [1-5] in trans dimensional random fields(TDRF):
> We have added a brief discussion on the relatedness of this line of work to our submission in the revision. While this line of work shares similar motivation of development of unnormalized energy models for sequences with our work, the similarities end there and this line of work is drastically different from our contributions as we describe below:
>
> -- \emph{Not related to masked language modelling:} Papers [1], [2] are not neural. [3],[4],[5] are neural but have very different parametrizations from MLMs and our approach.
>
> -- \emph{Focused on training:} These papers focus on explicit training of energy based language models, and sampling in [1], [2], and [3] is used for training and not for generation in the implementation and experiments. [4], [5] eschew sampling and focus on training via NCE. As discussed in the paper, training energy models explicitly is very expensive and noisy, as can be seen in the papers. In [3], it took 3 days to train over a corpus of 930K tokens, whereas large models like GPT3, BERT are trained on orders of magnitude more data. This renders these training approaches impractical for the amounts of data MLMs are typically trained on. Instead, our work focuses on interpreting large non-probabilistically pre-trained bidirectional models as energy models and proposes methods to generate sequences from them.
>
> -- \emph{No explicit focus on generation:} Tying to the point above, these papers do not perform and experiment with generation. The approaches are evaluated based upon perplexity and reranking capabilities. As discussed in our paper, [7] also performs reranking of ASR outputs based upon one of our energy parametrizations and achieves excellent results. We primarily focus on generating samples from masked language models.
>
> -- \emph{Differences in parametrization and sampling setup:} Although these works also use MCMC sampling (for training, not tested on generation), we would like to emphasize that MCMC and MH are general techniques to sample from unnormalized distributions and we do not claim to invent these techniques. Not only is the energy parametrization completely different in the neural TDRF [3], the proposal distribution is also different. [3] uses autoregressive proposal distribution whereas our work uses richer bidirectional MLM proposal distribution. A consequence of [3] using autoregressive models as proposal distribution is that it makes non-contiguous block sampling intractable. In contrast, we propose a valid MLM-based non-contiguous black MCMC sampling approach.
>
> In our paper and this part of the rebuttal, we have described how our work is different and novel compared to existing prior related work.
> In the next part, we respond to your other concerns.
>
> [1] B. Wang, Z. Ou, and Z. Tan, “Trans-dimensional random fields for language modeling,” ACL, 2015.
>
> [2] B. Wang, Z. Ou, and Z. Tan, “Learning trans-dimensional random fields with applications to language modeling,” IEEE transactions on pattern analysis and machine intelligence, 2018.
>
> [3] B. Wang and Z. Ou, “Language modeling with neural trans-dimensional random fields,” IEEE Automatic Speech Recognition and Understanding Workshop (ASRU), 2017.
>
> [4] B. Wang and Z. Ou, “Learning neural trans-dimensional random field language models with noise-contrastive estimation,” IEEE International Conference on Acoustics, Speech and Signal Processing (ICASSP), 2018.
>
> [5] B. Wang and Z. Ou, “Improved training of neural trans-dimensional random field language models with dynamic noise-contrastive estimation,” IEEE Spoken Language Technology Workshop (SLT), 2018.
>
> [6] Clark, Kevin, Minh-Thang Luong, Quoc V. Le, and Christopher D. Manning. "Pre-training transformers as energy-based cloze models." EMNLP 2020.
>
> [7] Salazar et al. "Masked Language Model Scoring", ACL 2020.

---

> > ### Author Response · Authors · 2021-11-18
> > **Response to cuUN (part 2/2)**
> >
> > In this part, we address your other concerns apart from the related work.
> >
> > We are primarily interested in masked language models and the energy networks that are implicitly trained via the MLM objective. While the general notion of MCMC and MH based sampling is well known, our major contribution is the specific concrete operationalization of these general methods for sampling from MLMs and extensive empirical investigation to answer several technical questions related to probabilistic interpretation of MLMs. We would like to emphasize the novel contributions of our work:
> >
> > -- MLMs can be cheaply trained on massive amounts of data and prior work has shown them to be excellent at learning representations of natural language. However, directly generating from these models has remained elusive, and our work focuses on this problem by interpreting them as implicitly training an energy model and generating samples from this model.
> >
> > -- Not only are we able to generate samples from MLMs, but our work also provides extensive evidence that non-probabilistic inexpensive methods to train good energy networks are possible and hopefully it will drive further research for such training approaches supplanting the typically expensive and unstable training methods for energy-based language models.
> >
> > -- As noted by reviewer dqpQ, this approach also paves way toward more application oriented generation by incorporating sequence level scores and signal into the energy function along with the MLM scores.
> >
> > -- As noted by you, we provide extensive empirical results on two different regimes of conditional and unconditional generation that shed light on various aspects of the nature of MLMs trained on large amounts of data.
> >
> > -- We also provide a theoretical discussion on estimation of free conditionals in fully connected neural language models and their appropriateness for training probabilistic energy models for text by contrasting our approach with prior work on probabilistic interpretation of MLMs.
> >
> > Regarding your comment about comparison with GPT: For the conditional generation task of machine translation (our experiments), GPT-based models are not prevalent. Autoregressive models are generally trained for MT and we have mentioned their performance in our original submission and the revision. For unconditional generation, since we focus on sampling from pretrained MLMs like BERT which are trained on different and smaller amount of data than GPT-2, we excluded this comparison. In fact, we use GPT-2 scores to assess the fluency of our generated samples in addition to human evaluation.
> >
> > Regarding your note on computational complexity: While we understand that sampling with our method is computationally more expensive than sampling from an autoregressive model, we would like to point out that we don’t require training or fine-tuning of any energy network. We use inexpensively pretrained MLM models on large amounts of data. Another benefit of our approach is potential speedup by parallelization and batching which is difficult to do with autoregressive models. Regarding practical usage, our approach is still practical for applications that require sampling edits to an already well-formed sentence using full bidirectional context and global features.
> >
> >
> > We have addressed your concerns regarding our work in this rebuttal and the revision. We hope that you would reconsider the novelty and significance of our contributions.

---

> > > ### Comment · Reviewer_cuUN · 2021-11-20
> > > **Re: Response to part 2/2**
> > >
> > > I understand that this work is primarily interested in interpreting non-probabilistically pre-trained MLMs as energy models and developing methods to generate sequences from them.
> > > I summarize and acknowledge the strengths/contributions of this work, as can be seen above from my first review.
> > >
> > > > For the conditional generation task of machine translation (our experiments), GPT-based models are not prevalent. Autoregressive models are generally trained for MT...
> > >
> > > This is confusing.
> > > It is said that for the conditional generation task (MT), GPT-based models are not prevalent. But then it is said that "Autoregressive models are generally trained for MT". Contradiction.
> > >
> > > >For unconditional generation, since we focus on sampling from pretrained MLMs like BERT which are trained on different and smaller amount of data than GPT-2, we excluded this comparison.
> > >
> > > Strictly, this is not a convincing feedback on comparison with GPT. You may try train GPT on the same data as in training BERT. With such comparison, the usefulness of the method developed in this paper for generation can then be fully justified.
> > >
> > > BTW, in Table 5, the results of Autoregressive (which, I guess, is GPT) are better than the proposed MH samplers. The descriptions about this Autoregressive model is not found in the paper.
> > >
> > > >Regarding your note on computational complexity...
> > >
> > > I understand what you said in this paragraph. But my concern "The proposed samplers seem to be computational very expensive... how to reducing the cost" is not addressed.

---

> > > > ### Author Response · Authors · 2021-11-23
> > > > **Response to cuUN part2/2 comments**
> > > >
> > > > Thanks a lot for your response. We clarify some of the concerns above.
> > > >
> > > > > This is confusing. It is said that for the conditional generation task (MT), GPT-based models are not prevalent. But then it is said that "Autoregressive models are generally trained for MT". Contradiction.
> > > >
> > > > There appears to be a miscommunication here. In our rebuttal, when we use the word GPT, we refer to the specific unconditional autoregressive model trained on large amounts of web data by the researchers at OpenAI. Regarding MT, we state in our paper and clarify here, that autoregressive approaches to train seq2seq MT models dominate the training landscape and non-autoregressive MT has seen little development in comparison to the autoregressive models. Despite this disadvantage, both MaskPredict (a nonprobabilistic non autoregressive MT approach) and our proposed approach, which samples from MaskPredict are able to perform competitively with autoregressive models for the task of machine translation. Another thing to note is that the main motivation behind experiments in Table 5 is to explore the modes of MLMs like the MaskPredict model with our parametrization and sampling scheme. Our annealing based approach is able to outperform MaskPredict showing that the modes under our proposed energy parametrizations do indeed correspond to sequences yielding good BLEU scores. The autoregressive results are provided as well for more context but are not necessary to answer the question if our parametrization’s modes correspond to MaskPredict’s predictions.
> > > >
> > > > > BTW, in Table 5, the results of Autoregressive (which, I guess, is GPT) are better than the proposed MH samplers. The descriptions about this Autoregressive model is not found in the paper.
> > > >
> > > > The autoregressive model is not GPT. It is a separately trained model on the parallel datasets used by the MaskPredict paper using standard cross entropy based training that is the default for training seq2seq MT models. Due to space constraints, we will add the details of this baseline to the appendix.
> > > >
> > > > > Strictly, this is not a convincing feedback on comparison with GPT. You may try train GPT on the same data as in training BERT. With such comparison, the usefulness of the method developed in this paper for generation can then be fully justified.
> > > >
> > > > Training a GPT sized autoregressive model on BERT’s data is computationally expensive and time consuming. For context, it took 4 days and 64 TPU chips to train BERT models. Training BERT models on larger amounts of GPT-2 data is likely more expensive and we would not be able to run such a baseline in the rebuttal window.
> > > >
> > > > However in the coming days, we propose to perform the following experiment which is somewhat unfair but would provide some context:
> > > > As seen in table 6, we report three metrics-- 2 human and one automatic. The automatic metric is GPT-2 perplexity which is not ideal but reasonable given that it was trained on much larger amounts of data than BERT. We will generate samples from the pretrained Huggingface GPT-2 model and report the GPT-2 perplexity on these samples, since human evaluation would take more time. Note that as mentioned in the rebuttal, this is not a fair comparison because GPT-2 was trained on larger amounts of data and GPT-2 perplexity would be biased toward generation from GPT-2 model.
> > > >
> > > > Regarding computational complexity: In our earlier response, we pointed to the fact that our approach is expensive but still practical for many applications as suggested by our experiments. Also, this approach is practical for edit based tasks which focus on perturbations to preexisting text. We agree that improving the computational complexity of this approach would be an exciting research direction for future work -- but we believe this is out-of-scope for the current work under review.

---

> > > > > ### Author Response · Authors · 2021-11-29
> > > > > **An update related to the small GPT-2 experiment described above**
> > > > >
> > > > > We generated sentences from GPT-2 under the same length,size settings as our other samples i.e. 120 random sentences of lengths sampled between 15-45. The avg GPT-2 perplexity on these samples is 115.8. This is slightly lower than the Bert-base raw model (125.4) shown in Table 6. As we mentioned above, this is a somewhat unfair comparison, and yet we find that the the samples from BERT-base are high quality and comparable to those generated from GPT-2 which was trained on different and larger data.
> > > > >
> > > > > Qualitatively, we observe that GPT-2 tends to give lower perplexities to longer sequences an also tends to give low perplexities to undesirable sequences containing a lot of repetition. This is a well-documented problem with GPT-2 generations (in prior work) and we did observe many high-scoring undesirable sequences among the 120 samples as well.

---

> > > > > > ### Comment · Reviewer_cuUN · 2021-12-01
> > > > > > **Re: An update related to the small GPT-2 experiment described above**
> > > > > >
> > > > > > Thanks for the updated experiment.
> > > > > >
> > > > > > >"we observe that GPT-2 tends to give lower perplexities to longer sequences an also tends to give low perplexities to undesirable sequences containing a lot of repetition. "
> > > > > >
> > > > > > It would be better if such sample generations can be added, as along with the new perplexity results of GPT-2 and your model.

---

> > > > > > > ### Author Response · Authors · 2021-12-02
> > > > > > > **Re:**
> > > > > > >
> > > > > > > Thanks for the suggestion. We will definitely add some such samples from GPT-2 in the appendix alongside the section containing samples from BERT E_raw using our approach.

---

> > ### Comment · Reviewer_cuUN · 2021-11-20
> > **Re: Response to part 1/2**
> >
> > Thanks for the response from the author(s).
> >
> > It is good to see that the author(s) expand the differences between [1-6] and this work in the feedback, which, however, may not be sufficient. The point is: being clearly positioned in the literature and properly claiming contribution with respect to prior works is one of the key questions in reviewing a paper.
> >
> > I suggest the author(s) to clarify this work from the perspectives of model definition, model learning, and model usage, rather than mixing things, which may confuse the readers.
> > First, there are prior works on energy-based LMs (e.g., [1-5] and [Deng et al., 2020]), yet the definition of the energy-based LMs from non-probabilistically pre-trained MLMs (raw scoring and locally normalized scoring) in this work is somewhat novel, namely using different energy parametrization.
> >
> > Second, this work is not related to learning energy-based LMs, and in fact it neither improves nor evaluates model learning. It focuses on sampling/generation given the model, i.e., proper sampling from pre-trained MLMs, which itself is a meaningful topic. Thus, the comments in Related work on learning energy-based LMs, which are biased towards the superiority of the MLM objectives, are neither very necessary nor justifiable and in fact out of the scope of this paper. In fact, it is unclear from the literature which is better when evaluated in a common experimental setup.
> >
> > Third, since this paper focuses on sampling/generation given the energy-based LM, connecting and comparing to prior sampling methods from energy-based LMs are clearly needed.
> > Note that model learning often involves model sampling, so instead of emphasizing the difference between [3] (which addresses model learning) and this work (which addresses model sampling), the Related work section should comment on the differences about model sampling in both works. Both are based on MH but applied to different models and thus with different proposals. Good to see the authors have some such comments in \emph{Differences in parametrization and sampling setup:} in the feedback.
> >
> > Hope the above clarifies my main concern about this paper when I give the current recommendation, which is still not well addressed in the updated paper.
> > The response from the author(s) contain some good clarification.
> > I suggest to polish the main text, particularly the Related work section. I'm happy to adjust the score if the authors take efforts to improve the paper.

---

> > > ### Author Response · Authors · 2021-11-23
> > > **Response to cuUN part1/2 comments**
> > >
> > > Thanks for your response to our rebuttal. We have uploaded a new revision, in which we have cut down on some exposition to make space for more detailed and clearer related work and introduction sections. We have especially described how prior work has mostly focused on training and how the parametrizations considered in prior work are different from our work. We explicitly highlight that our work doesn’t focus on training energy-based networks and instead we are mainly interested in sampling from non-probabilistic MLMs by interpreting them as implicitly training energy networks, which we view as computationally advantageous.
> > >
> > > Thanks a lot for additional feedback on your impression of our positioning of the work in the context of prior work. We do not intend to downplay the importance of training energy networks from scratch over sampling from MLMs. In fact, we hope that our findings in this work promote future work on explicitly training efficient energy networks for text with indirect objectives like MLMs’ objectives. We have elaborated on this aspect in the related work section and have more explicitly connected our work to prior training-focused work on energy networks.
> > >
> > > Regarding comparison to the sampling methods used for training energy networks in prior work: We have reflected this discussion in our related work as per your advice. You are correct that approaches like [3] which train energy networks using MCMC sampling methods are amenable for generation using the same sampler. Therefore, we have discussed the differences between [3] and our approach in greater detail. Mainly [3] uses a completely different energy architecture and an autoregressive proposal distribution, while our approach uses MLM based energy networks and proposal distribution. Also, [3] focuses on training energy models from scratch while we aim to sample from pretrained non-probabilistic representation learning models. The very important difference is the proposal distribution in [3] and our work. [3] uses unidirectional autoregressive models for proposals while we use bidirectional MLM contextual representations for our proposal distribution. One clear consequence of this is that we are able to perform non-contiguous block sampling with our approach which is non-trivial to do with an autoregressive proposal.
> > >
> > > We would be happy to incorporate more suggestions regarding elaboration of the related work section.

---

> > > > ### Comment · Reviewer_cuUN · 2021-12-01
> > > > **Re:**
> > > >
> > > > Thanks for the feedback. The paper has been improved, although the added discussions updated in the related work section are not as clear as the authors' clarifications in the feedback above. Better clarification in the main text could be made.

---

> > > > > ### Author Response · Authors · 2021-12-02
> > > > > **Re:**
> > > > >
> > > > > Thanks for your feedback and consideration. The space constraint makes it difficult to discuss various aspects of the paper in as much detail as we would ideally like to. In future revisions, we will try to work around the constraints to make the discussion about related work more elaborate.

---

### Official Review · Reviewer_zwFH · 2021-11-02

**Correctness:** 4
**Technical Novelty And Significance:** 3
**Empirical Novelty And Significance:** 3
**Recommendation:** 6
**Confidence:** 4

**Main Review:**

strengths：
1. The author introduces energy networks to rectify the incorrect assumption which is existed in previous work [1].
2. This paper proposed two novel energy parametrizations and introduces Metropolis-Hasting sampling to text generation based on MLM models.
3. All methods are empirically verified in a variety of experiment settings.

weaknesses：
1. This paper lacks the introduces of previous works, such as undirected generation approaches and energy networks, which makes the paper is hard to understand without reading the references.
2. There are many experiments between proposed method with the degenerate Gibbs sampling, but do not compared with other generation models like GPT, BART, etc. The potentiality of  proposed methods can not be estimated.
3. The computational complexity of MH sampler is very high as described in this paper, which may not valuable for real applications.

questions:
1. Is the methods proposed in this paper applied in inference stage? Which means the model cannot applied in other datasets except the dataset which is used for training the pre-trained model.

typos:
In section 4,  "We empirically study the proposed Metropolis Hastings scheme .... and the task of uncondtional generation." -> unconditional

Reference:
[1] Wang and K. Cho. BERT has a mouth, and it must speak: BERT as a Markov random field language model. arXiv preprint arXiv:1902.04094, 2019.


**Summary Of The Paper:**

The paper tries to interpret masked language models as energy-based sequence models and proposes two energy parametrizations derivable from the trained MLMs. The primary contribution in paper is to rectify the incorrect assumption in the prior work and propose Metropolis-Hastings (MH) based sampling algorithms for these energy networks. And another contribution is to design a block-replacement proposal distribution for improve mixing of the Markov chain in our proposed MH sampling framework, which results in faster generation and better samples. The paper justifies their findings on the conditional generation task of neural machine translation (NMT) by the pre-trained models MASK-PREDICT and the task of unconditional generation by Bert.


**Summary Of The Review:**

This paper interprets the masked language model with a novel perspective, and introduces  energy networks and Metropolis-Hasting sampling to text generation based on MLM models. It proves the effectiveness of MLM models on text generation, a good paper to accept.

---

> ### Author Response · Authors · 2021-11-18
> **Response to zwFH**
>
> Thanks for engaging with our work and your insightful review.
> We have addressed the presentation concerns in the new revision and have added more detailed introduction to the methods.  Although we would like to make our exposition more verbose, the page limit makes this difficult. We are happy to take additional comments of yours on the revised draft.
>
> Regarding your note on comparison with GPT: Our work focuses on sampling from large pretrained MLMs like BERT, which are trained on different and smaller amounts of data than GPT-2, therefore the generation comparison would be unfair. However, we did some preliminary experiments for debugging our approach which involved treating autoregressive models like GPT-2 as the energy models and running our MH sampler with MLM proposals with the autoregressive energy. The generated samples were decent but this is clearly an inferior sampling strategy to the exact ancestral sampling strategy for the autoregressive models. The MH strategy is further affected by the divergence between the MLM proposal distribution and the GPT2 energy. We will add this in the Appendix of the forthcoming revision in a few days.
>
> Regarding computational complexity: While we agree that our method is more computationally expensive than ancestral sampling for autoregressive models, faster sampling methods from MLMs are an open research question. Regarding practical usage, our approach is still practical for applications that require sampling edits to an already well-formed sentence using full bidirectional context and global features.
>
> Moreover, an attractive feature of our approach is that we do not require training or fine-tuning of an energy model and instead focus on sampling from inexpensively trained MLMs on large amounts of data.
>
> Another upside to our approach is more straightforward ability to parallelize generation compared to autoregressive generation.
>
> Regarding your question: We aim to generate samples from pretrained masked language models. For models like BERT which are trained on a large amount of data spanning various domains, our approach should be able to generate domain specific sentences fairly easily without any finetuning.

---

### Official Review · Reviewer_dvua · 2021-11-03

**Correctness:** 4
**Technical Novelty And Significance:** 3
**Empirical Novelty And Significance:** 4
**Recommendation:** 8
**Confidence:** 4

**Main Review:**

The strengths of the paper is its technical novelty and rigor as well as its thorough experimental studies. The weakness of this paper is its lack of clarity on a few specifics.

The proposed method could approximate energy-based sequence models with a good proposer (i.e., MLM), and the paper clearly discusses its general theoretical advantages over a Gibbs sampling alternative. The extensive experiments on machine translation and unconditional generation well support the claims.

However, I would appreciate more details on the specific target distributions used in the experiments: e.g., I am not familiar with MASK-PREDICT and it is not obvious to me how the general technical remarks apply to this specific case.


**Summary Of The Paper:**

The paper proposes to use metropolis-hasting Monte-Carlo to draw samples from energy-based distributions over sequences. They construct proposal distributions based on trained masked language models (MLMs). Compared to an alternative that uses Gibbs sampling on MLMs, the proposed method is theoretically sound (i.e., correctly approximating the target distribution) and empirically effective on a couple of sentence generation tasks.

The proposed method is novel and principled.


**Summary Of The Review:**

Overall, I think it is a good paper. I believe my concerns are more about presentation instead of the actual methods.

---

> ### Author Response · Authors · 2021-11-18
> **Response to dvua**
>
> Thanks for engaging with our work and your positive assessment of our work. We have tried to improve the presentation of our methods and also expanded upon the description of the related work in the revision. Although we would like to make our exposition more verbose, the page limit makes this difficult. We are happy to take additional comments of yours on the revised draft.

---

### Official Review · Reviewer_dqpQ · 2021-11-03

**Correctness:** 4
**Technical Novelty And Significance:** 4
**Empirical Novelty And Significance:** 3
**Recommendation:** 8
**Confidence:** 5

**Main Review:**

*Pros*
- The paper tackles an interesting problem of sampling from energy functions with local conditionals.
- The theoretical contributions of the paper are quite sound. The rationale behind (wang and cho'19) faulty assumption and the use of MH as an alternative.
- The experiments show interesting trials with different variations and honest explanations of the obtained scores. Overall sampling from MLM is a new task compared to auto-regressive LMs it is natural that the performance obtained is going to be slightly inferior.

*Areas of enhancement*

- The readability of Section 2 (Energy) could be enhanced, the connections between E_local and E_raw could be elaborated since both are quite similar computationally. The current structure gives the impression that they are completely different formulations.

- Section 6 (Experiments) could be enhanced by providing a global narrative and coherence between the experiments, It is easy to get lost and compare different tables. For example, it took me some time to notice that the results from Table 4 are superior to those of table 3 and 2 and therefore Block MH sampling is useful.

- For Open-ended text generation experiments the corpus level diversity using (SELF-BLEU) should be reported overall, some generation examples could be displayed in the appendix to judge the overall quality (see questions below).


*Questions / recommendations to authors*

- What are the theoretical/conceptual differences between E_norm and E_raw? I see they have been used both in previous work and empirically they perform differently, are there any conceptual differences that motivated you to use both formulations? If it was merely experimental it is fine just to mention it clearly in the text.

- Samples from MH are known to have high auto-correlation (i.e. i.i.d), this is due to the nature of the local proposal. For applications like open-ended generation (unlike NMT) where the user expects large corpus diversity, this is a crucial feature to have. Although authors report *Novel* transition rates this could be misleading as it only shows the novel transition while the space of effective X could still be limited. I recommend reporting SELF-BLEU in the open-ended generation experiments to indicate corpus-level diversity. For more details see: https://arxiv.org/pdf/1811.02549.pdf

- section 7: "Although the autoregressive approach is superior (30.18 de-en ..."  it would be great adding those results in table 5 to allow easier comparison even when they are superior to non-autoregressive models (which is understandable).

- Generation examples: For transparency, one would like to see examples generated from the MH algorithm, perhaps dump a subset of all generations of each method in a table in the appendix. One would like to asses the auto-correlation effect of MH on the diversity of the generated samples.

- The locally normalized scoring formulation is referred to as  $E_local$ $E_norm$, "norm" and "local" interchangeably this confuses the reader.

- Equation of $E_{local}$ in the bottom of page 3 has two symbols t and i. it is not clear where the $i$ comes from is this a typo?

- Page 3: "the following free conditionals are inconsistent" in fact this example is not clear, I had to go back and forth many times with the appendix to get it right. It would be great to take the time to explain this example in a self-contained way.

- Figure 2 is not clear that is the green curve

**Summary Of The Paper:**

This work makes a step towards sequence generation from MLMs such as BERT. This is an interesting task itself but also managing to do so will enable more applications about sampling from Energy functions of text, think of combining MLMs scores with sequence level scores obtained by the same MLM and using the resultant energy to obtain samples directly.

(Wang and Cho 19) proposes an energy parametrization similar to E_raw in this paper for a Gibbs sampling approach under a faulty assumption that the token conditionals are independent from each other this has been clarified by Wang and cho'19 later in an independent post.  In this work, the authors extend on this clarification both theoretically and empirically. in section "conditional distribution under E_raw" They show that MLM conditionals do not correspond to the global sequence distribution specified by E_{raw}. And if one wanted to do Gibbs sampling, they drive the correct conditionals under E_{raw}, however, these are expensive to compute as they will need |V| times fwd passes of the MLM to compute the conditional for each position $t$.

An alternative approach proposed by the authors is to use Metropolis Hasting in brief they use two scoring functions,

1) $E_{raw}$ raw scoring (sum of MLM logits) the one proposed by (Wang and Cho 2019)
2) $E_{norm}$ locally normalized scoring (sum of MLM softmax outputs).

Both of those parametrizations are not globally normalized and are hard to directly sample from. Therefore they propose to use MH to sample from those scorers using a proposal distribution for the transition probability $q(X',X)$.

One advantage of MH is that q here could be any proposal distribution authors decide to use here to use the Masked Conditionals as a proposal distribution. Additionally, they try variants of this proposal distribution q,  by varying temp, nucleus sampling, and block MH sampling, the latter is motivated by the low mixing rates of the original proposal q despite its high acceptance rate.

This work conducts several experiments to show the following:

1- Comparing MH with E_raw and E_norm vs the faulty Gibbs Sampling approach. Degenerations are obvious in the case of the Gibbs sampling which conforms to the theoretical motivations described above.

2- Temperature / Nucleus sampling: the main findings here: is that the high rejection rates of MH samplers and that gibbs sampler could be useful in practice with low temperatures.

3- Block MH sampling with the proposal q: showing improvements than the previous proposals in terms of novel transition rates and BLEU scores for NMT experiments.

4- extra experiments with temperature annealing of the energy function comparing with Mask-predict and autoregressive LM

5 - human evaluation for open-ended generation


**Summary Of The Review:**

Overall the paper is theoretically sound experiments are sufficient in my opinion and shows that the proposed method works.
I would love the authors to enhance the accessibility/readability of the manuscript by unifying mathematical notation, defining the rationale behind using $E_{local}$ and $E_{norm}$ rather than long paragraphs referring to previous work instead of motivation. It would be great to simplify the connections between them in layman terms (sum of scores and sum of softmax outputs).

---

> ### Author Response · Authors · 2021-11-18
> **Response to dqpQ**
>
> Thanks a lot for a detailed review and positive assessment of our work. We have incorporated your suggestions and improved the presentation and readability in the new revision. We modified some sections like introduction to make them shorter in order to make the paper clearer. Although we would like to make our exposition more verbose, the page limit makes this difficult. We are happy to take additional comments of yours on the revised draft.
>
> We have also included 10 samples with E_{raw} and they form a diverse set. We are generating new samples so that we can compute self-BLEU and report them in the paper. This will be reflected in future revision in a few days.
>
> Regarding your questions:
>
> -- We used the two formulations because they are related and prior work around MLMs has considered them in some form. We believe E_{norm} is slightly more constrained with regards to generating samples because of the normalization constraint, but the decision was largely experimental and based on precedent. We have explained this in the revision.
>
> -- You are correct to point out that autocorrelation is a concern with MCMC methods for text like the proposed scheme. In our early experiments, we chose interleaved samples with generous gaps in the chain to mitigate the issue of autocorrelation. Hoever, since this paper is a study of the energy model behind the MLM objective, we took a more computationally expensive approach of running a different chain for each sample we report our results on. This results in a diverse set of samples, which can be seen in the shared samples in the revision. It also rules out any direct effects of autocorrelation with our proposed scheme. We will add self-BLEU scores in the upcoming revision soon.
>
> -- The green curve in figure 2 is the negative of E_{local}. This is to show that annealing causes this quantity to grow steadily with epochs.
>
> -- We have made changes regarding the presentation and exposition in our revision.

---

### Decision · Program_Chairs · 2022-01-20

**Decision:**

Accept (Poster)

**Comment:**

This paper interprets pre-trained masked language models (MLMs) as energy-based sequence models and designs a tractable MCMC sampling algorithm based on Metropolis-Hastings with proposals derived from MLMs themselves.

The strategy is simple, reasonably elegant, and fixes technical mistakes of prior work. The proposed algorithm addresses intractabilities of some naive MCMC schemes, does not require modifications to MLM training, and makes good use of MLMs themselves as proposals thus being crucially economical about resources.

We had some concerns about speed of generation and the paper's positioning with regards to existing strategies for sampling from energy-based models (already during parameter estimation). While I understand that for many applications speed of generation is crucial, I think that on its own should not keep this line of research outside our best venues. And I hope steps like this one will lead to faster algorithms in the near future. I do relate to the issue of positioning, and I am glad the authors did not take it lightly. In the rebuttal phase the related work and positioning have been improved, but the authors remarked that the limited space for the camera-ready was preventing them from expanding the discussion. A note to authors: it's not a bad idea to have an expanded related work section in appendix.